# Non-Communicable Diseases and Transitioning Health System in the Democratic People’s Republic of Korea during COVID-19 Lockdown

**DOI:** 10.3390/healthcare10102095

**Published:** 2022-10-20

**Authors:** Jin-Won Noh, Kyoung-Beom Kim, Ha-Eun Jang, Min-Hee Heo, Young-Jin Kim, Jiho Cha

**Affiliations:** 1Division of Health Administration, College of Software and Digital Health Care Convergence, Yonsei University, Wonju 26493, Korea; 2Industry-Academic Cooperation Foundation, Yonsei University, Wonju 26493, Korea; 3Department of Health Care Management, College of Bio Convergence, Eulji University, Seongnam 34824, Korea; 4Department of Health Administration, Yonsei University Graduate School, Wonju 26493, Korea; 5Department of Health Administration, Dankook University, Cheonan 31116, Korea; 6Moonsoul Graduate School of Future Strategy, KAIST, Daejeon 34141, Korea

**Keywords:** Democratic People’s Republic of Korea (DPRK), COVID-19, non-communicable disease (NCD), health inequality, scoping review

## Abstract

While there are increasing concerns on COVID-19 situation in Democratic People’s Republic of Korea (DPRK, or North Korea), little is known about North Korea’s health system function for Non-Communicable Diseases. Given the scarcity of available evidence, a scoping review was conducted in peer review articles from MEDLINE, SCOPUS, and Web of Science, and policy literatures from *Rodongshinmun,* state-run media in North Korea to analyze the North Korea health system and COVID-19 pandemic. The transition to a market economy is expected to deepen the gap between the rich and the poor over access to health care, causing a new type of health inequality in North Korea. COVID-19 lockdown intensified the DPRK’s economic predicament exacerbating shortage of health financing on non-communicable diseases. The case study of mixed evidence from scoping review indicates that NCDs prevention and management are not functional in the transitioning health system under chronic economic crisis and isolation. This study indicates that NCDs prevention and management are not functional in the transitioning health system under chronic economic crisis and isolation. The destabilized markets under COVID-19 lockdown intensified the DPRK’s economic predicament and exacerbated the chronic shortage of health financing especially to NCDs.

## 1. Introduction

Non-Communicable Diseases are not only the leading cause of more than 70% of all deaths worldwide [1], but also attributing cause of COVID-19 mortality. NCDs are major risk factors for COVID-19 patients [2]. According to studies in Wuhan, China, 96.2% of those dying of COVID-19 had non-communicable disease (NCD) comorbidities such as hypertension, diabetes, ischemic heart disease, and cancer [3]. Both the severity of COVID-19 and its association with NCDs have been reported in Spain, China, and the United States, suggesting that these tendencies are not limited to a few countries [4,5,6,7]. All of which imply continuous needs for primary healthcare system to manage NCDs even in COVID-19 situation [8]. The combination of ageing population, elderly with multiple chronic diseases and frailty were associated with COVID-19 infection and higher severity requiring hospitalization, intensive care [9].

Democratic People’s Republic of Korea (DPRK, or North Korea) has the highest proportion of the elderly population compared with other Low- and Middle-income countries in Southeast Asia region. Life expectancy has rebounded from 65 in the peak famine years of 1995–1998 to 72 in 2017, with 13.4% of the population now over 60 [10].

While there are increasing concerns on COVID-19 situation in North Korea, little is known about non-communicable disease and the health system for NCD managements despite considerable NCD risk factors in North Korea. And the current state of North Koreans health and diseases information is limited access. Given the scarcity of available literatures in North Korea, this study conducted a scoping review of peer review articles and policy related literatures through state-run newspaper to analyze the NCDs in North Korea in particular consideration of health system impacts of COVID-19 pandemic. While we fully recognize the methodological limitation from lack of available dataset, this study aims to indirectly provide North Korea information on the status of NCDs and health system in COVID-19 pandemic. This scoping review can give an overview of accessible policy evidence for more comprehensive understanding of NCDs and health system dysfunction in North Korea under the COVID-19 pandemic.

## 2. Materials and Methods

A scoping review was conducted on peer-reviewed journals in English and then mixed with policy and media documents from North Korea’s state-run media to provide comprehensive evidence on NCDs and health policy (health policy and health system). All relevant documents published in the English and Korean language that presented NCDs and healthcare in North Korea were included. Literature focusing on North Korean refugee health were excluded. The main sources of searching peer review articles were MEDLINE, SCOPUS, and Web of Science, which are representative web-based electronic databases. A systematic literature search was performed in November 2020. Articles chosen were published between 1 January 2010, and 10 November 2020, and the date of document extraction according to selection criteria. The inclusion criteria comprised original papers in the English language only. The media analysis was conducted with Rodong Sinmun, official newspaper of the Central Committee of the Workers’ Party of Korea, which is regarded as a formal source of North Korean government’ policy and viewpoints on domestic and international issues. The media searching was done in articles and documents published in the first year of COVID-19 pandemic in North Korea between 22 January 2020, and 1 December 2020. A search strategy for each target data source was established by selection criteria in the form of free text and controlled vocabulary for each combination of key concepts (Appendix A). On this basis, by using filtering and qualifying functions provided for each data source, Boolean operator, and logical operator, literature matching the target was effectively extracted. Since the primary research on the DPRK is suspected to be insufficient, we searched for a broader range of NCDs to maximize sensitivity.

We searched a total of 243 papers (117 Medline, 75 Scopus, and 51 Web of Science) that were potentially relevant to the study subject. Duplicates were removed based on the thesis name, author, year, and journal name using EndNote (X20), and 153 papers were selected. As a result of this selection, titles and abstracts of 118 papers that had been screened were excluded, as they did not meet the inclusion criteria. After the remaining 35 papers had been subjected to full-text screening, 6 papers were ultimately reviewed (Figure 1 and Table 1). Between 22 January 2020, and 1 December 2020, Rodong Sinmun continuously issued policy related to health and COVID-19 pandemic. The media analysis searched a total of 491 articles including 402 articles related to health including COVID-19 and non-communicable disease and 88 articles related to health policy, law, and health system.

## 3. Results

### 3.1. Transitioning Health System with Financing Failure

North Korea’s health care system shares many characteristics of health system transitions in other socialist countries in 1990s. North Korean health facilities were operated through a centrally raised health revenue within a centralized administrative system. The DPRK authorities attempted to provide comprehensive free health care through the formal health care system [11,12]. It was intended to promote public health, prevent diseases, and increase the effectiveness of treatment through the integration of traditional and western medicine [13].

As the economic crisis accelerated in last two decades, the socialist health care system was scaled back [12,14]. North Korea theoretically pursued universal health coverage, but the actual health coverage rate is low due to shortage of health financing. It was difficult for the North Korean government to secure a budget to support the centralized healthcare system. The health system has been significantly weakened due to the chronic shortage of health financing. Primary health clinics and hospitals were chronically short of the budgets and necessary supply of medicine and equipment. As a result of analyzing two census data in 1993 and 2008, life expectancy decreased significantly, and mortality in infants and the elderly increased.

The DPRK has reached the limits of what free health care under the socialist system can provide, and accordingly, has provided health care services through both formal and informal channels [11]. As the informal market mechanism expanded rapidly, commercial distribution of essential items was possible outside the formal economic system. Although socialist healthcare system partially provided the medicine and medical equipment without charge, more patients directly obtained medicine through market and private health care services, as de facto admitted by North Korean authorities [12]. The expansion of the informal market, the degenerate nature of socialist health systems, and lack of medicine and other health care supplies have increased the self-treatment and traditional medicine usage of North Koreans [11]. The informal market has more of an impact on the decline in the development of the formal market because of relatively low trust and expectations on the part of patients and acceptance of the informal market [12]. There was an increasing gap between universal healthcare agenda and economic realities.

### 3.2. NCDs and Changing Health Disparities

In the transitioning health system, NCDs were a considerable health care burden on the DPRK comparable to widespread malnutrition and infectious disease [15]. The burden of NCDs is high accounting for 79% of all deaths in North Korea. Half of these burdens are related to cardiovascular disease. According to the WHO, it was estimated that the age-standardized mortality rate of NCDs in North Korea increased from 607 to 677 in the period from 2000 to 2016 [8]. This is very high compared to 267 deaths per 100,000 in South Korean population having ethnic similarity. According to the major causes of death in North Korea in 2017, NCD mortality of cancer and cardiovascular disease were higher than the mortality rate from infectious diseases where infectious diseases accounted for 4.8%, cancer 17.7%, and cardiovascular disease 38.4% [16].

**Table 1 healthcare-10-02095-t001:** Brief Summary of Reviewed Papers.

Reference	First Author (Published Year)	Study Population	Type of Study	Study Purpose	Main Results	Theme		
						NCDs Management Status and Policies Change	Health Inequality	Changes in Health Care System
[14]	Lee (2013)	Not applicable	Road map	To establish integrated health care system of Korea Peninsula	Establishment of an integrated healthcare system execution plan over four phases		V	V
[17]	Tran (2011)	200 women in 6 provinces of DPRK	Cross-sectional study	To measure the level of knowledge and perception of women about cervical cancer	No significant difference in knowledge between urban and rural areas about cervical cancer Most women do not know that it is preventable	V	V	
[11]	Lee (2020)	383 North Korean refugees	Cross-sectional study	To find socio-economic and political determinants of ill health and healthcare access in North Korea.	Large disparities in health and access to healthcare in North Korea, associated with political and economic inequalities.		V	V
[15]	Lee (2013)	National statistics of North Korea derived from WHO	Cross-sectional study	To evaluate the overall current disease burden of North Korea through the recent databases	Burden from deaths caused by NCD is greater than from that caused by CD or malnutrition	V		
[12]	Soh (2016)	19 North Korean refugees	Cross-sectional study	To examine the emergence of informal healthcare practices in North Korea	The informal healthcare system complements the formal sector, which does not mean the end of the formal socialist system.			V
[13]	Canaway (2017)	Not applicable	Field notes and photos	To impart reflections on North Korea’s healthcare system	The integration of ‘koryo’ and modern healthcare in the fields of education and medicine was in practice			V

The increase in adult mortality reflects the deterioration of living standards and health system transition in North Korea. As the formal health sector was scaled back, the DPRK does not have sufficient resources to adequately respond to NCDs prevention. For example, the DPRK’s policy on cervical cancer emphasizes the importance of preventive screening [17]. However North Korean women rarely receive HPV vaccines to prevent cervical cancer [17]. Women in rural and urban areas lack knowledge of the symptoms, causes, and potential for preventing cervical cancer [17], and health barriers exist because of high health care expenses and insufficient medicine [11]. Women and children faced particular health threats [14]. Child mortality in the DPRK was found to be more attributable to NCDs than to communicable diseases (CDs) and malnutrition [15].

In addition, the transition to a market economy is expected to deepen the gap between the rich and the poor over access to health care, causing a new type of health inequality in North Korea. Along with the expansion of the market economy, household incomes were largely relying on the informal market activity. Economically marginalized groups from emerging economy lack resources to access medical services in the private health market as the informal out of pocket money on healthcare services were especially required for medicines. This uneven impacts of expanded private health markets were resulted in aggravating inequality of accessing health care services for NCD burdens of the marginalized population [11]. It is difficult to access healthcare service in the formal and private health sector, respectively, for groups politically alienated from the existing socialist system and groups alienated from the emerging market economy.

### 3.3. Impacts of COVID-19 Lockdown

North Korea have closed its border since January 2020 and reported that there are no confirmed COVID-19 cases. As the pandemic continues, the provision of health care in the pandemic affected countries has deteriorated [9] under combined impacts of social distancing and lockdown, economic crisis, and health system disruption. In the fragile countries without resilient economic system, the border closure has a risk to bring economic devastation leading to additional barriers to food and health care service [18]. Given its economic instability, it seems quite clear that COVID-19 lockdown has affected the healthcare access to NCDs in North Korea.

While many other countries hesitated to close their borders early in the COVID-19 pandemic with concerns on negative economic impact, the DPRK government has created unprecedented border closures that have halted trade between the DPRK and China, exacerbating economic poverty [19]. Border closure and lockdown destabilized the price of essential medicine and goods in the market. There were 90% decreased in food import from China, compared with those in 2019 and price of medicine from China was increased. The formal or informal business in the market was extremely reduced with social distancing and lockdown. The DPRK’s economic crisis is now intensifying due to COVID-19 lockdown, which exacerbate the chronic shortage of health financing [20]. Lack of resources in the North Korean health care system limits the provision of health care services. Economic hardship has prevented NCD prevention and management in formal health system from functioning [21].

Economic inequality among North Koreans intensified in the process of transitioning to an unstable informal economic system. In the aggravating market function and livelihood in COVID-19 pandemic, financial barriers to private health sector were much worsened. Public health measures used to prevent the spread of COVID-19, such as lockdown and social distancing, have limit people’s activity, which can aggravate livelihood causing financial barriers to health services [18]. In COVID-19 pandemic, this is considered to aggravate healthcare barriers associated with NCDs, a chronic health burdens in North Korean society [16].

Therefore, the impacts of COVID-19, such as economic deterioration, border closure, and social distancing, disrupt the supply of primary health care in the case of NCDs, leading to health inequalities [9]. As the pandemic continued, low and lower middle-income countries are increasingly under dual burden of diseases for which integrative approach between the COVID-19 response and NCD management is required even in poor resource settings.

## 4. Discussion

This study presents the first case study of the North Korean health care system in relation to NCDs through the mixed methods scoping review of the existing peer review articles on NCDs and related policies, health inequalities, and transition of the health care system as well as articles of state-run newspaper which are key sources of policy announcement in North Korea. Before the COVID-19 pandemic, supply chain and healthcare system in North Korea were more weakened due to strictly enforced international sanctions and reduced trade with China. The limited financing health-related endeavors were reported due to unstable economic situations [22]. However, North Korea has increased burden of NCDs poor health status, malnutrition, and communicable diseases [23]. North Korea has imposed strict nationwide lockdown due to the COVID-19 pandemic. And they have been faced humanitarian emergency due to chronic food insecurity and limited access to basic services, accelerated by declining imports of medicines and food, affecting health care [24]. Therefore, overview of accessible policy evidence for more comprehensive understanding of North Korea health status and health system under the COVID-19 pandemic is needed. Our study shows that health care in the DPRK was provided through both formal and informal health systems resulting in potential financial barriers to access NCD cares. To our knowledge, much remains to be discussed regarding the health system impact of COVID-19 preventive measures on NCD health care provision. According to previous study, NCDs patient living in LMICs having large NCD burden shown limited access to services for their health conditions. People with NCDs lives were worse due to partially or completely degraded healthcare services for NCDs and related complications in pandemic situations and decreased accessibility to basic medicines [25,26]. However, the following limitations should be considered in this study. First, despite its effort to access policy related literature from the state-run media, formal policy documents were hard to be accessible outside North Korea. Thus, this review lacks more direct policy documents related to NCDs and COVID-19 pandemic. In addition, similarly, the limited information available from the DPRK has limited the latest research on the DPRK’s health care conditions. Further research is needed to address the current evidence on COVID-19 status and the latest policy trends in the DPRK. In addition, more accurate estimates of the burden imposed by NCDs and an analysis of the NCD care system may contribute to future gains in North Korean society.

## 5. Conclusions

The DPRK has closed the borders, minimizing public damage from pandemic [18], and claimed that it had no confirmed cases of COVID-19 [27]. The COVID-19 pandemic looks successfully controlled. However, the evidence from scoping review implies that the destabilized markets under COVID-19 lockdown intensified the DPRK’s economic predicament and exacerbated the chronic shortage of health financing especially to NCDs [20]. Considering that the existing socialist health care system does not function well, and many North Koreans partially rely on the private health care system [28], the contraction of the informal market, and the increase in poverty due to the COVID-19 pandemic weaken access to overall health care services. Given the limited health care resources in the DPRK, it can be assumed that the COVID-19 pandemic has made it difficult to distribute health care resources equitably, resulting in health inequity.

The present study revealed that health care provided through both formal and informal health system in the DPRK has caused potential financial barriers to access NCD cares under the COVID-19 pandemic. This study demonstrated the double burden of diseases which caused by existing malnutrition and communicable diseases and increasing NCDs before the COVID-19 outbreak, and the healthcare system aggravated by the COVID-19 pandemic in North Korea. This study was meaningful in providing baseline data on DPRK health status and healthcare system. These baseline data could support digital technologies such as COVID-19 planning to use blockchain technology [29]. These reliable and adequate baseline data can contribute to well versed decisions in COVID-19 pandemic situations. In addition, these baseline data can support several international humanitarian organizations to reduce disparities in health and accessibility to healthcare in North Korea.

## Figures and Tables

**Figure 1 healthcare-10-02095-f001:**
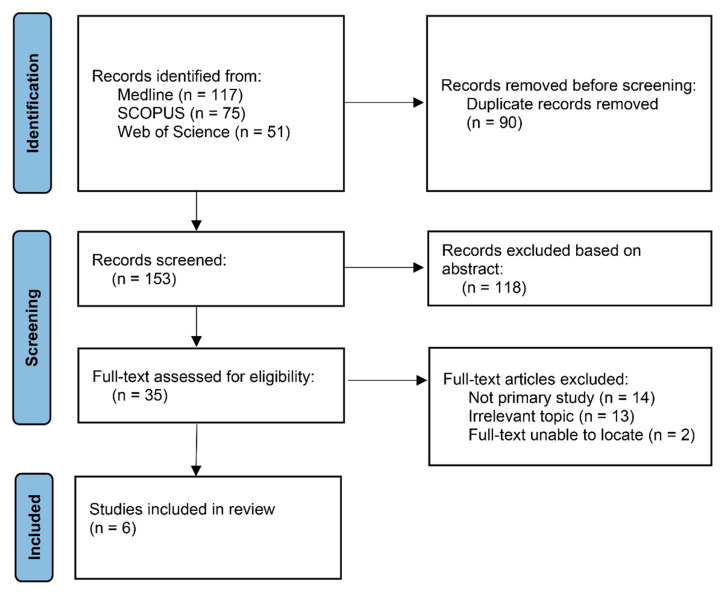
Flow Diagram of Study Selection.

## Data Availability

The datasets used and/or analyzed during the current study are available from the corresponding author on reasonable request.

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
