# Peer review of "Non-Communicable Diseases and Transitioning Health System in the Democratic People’s Republic of Korea during COVID-19 Lockdown"

_healthcare, 2022, doi:10.3390/healthcare10102095_

Round 1

Reviewer 1 Report

The subject addressed is particularly interesting, but the article requires a series of improvements for publication:

- in order to have a correct picture related to the situation of non-communicable diseases, a detailed information related to their situation is necessary even before the pandemic;

- an analysis of the situation of non-communicable diseases during the pandemic is required compared to the previous situation.

Discussions are insufficient, it is necessary to make an analysis of the situation of non-communicable diseases from the pandemic period and with those from neighboring states and even with states with both similar and different policies, obviously studying the data published in specialized journals.

The conclusions require an improvement and an emphasis on the relevance of the study for the proposed subject.

Author Response

October 12, 2022

Healthcare (healthcare-1966962)

Non-communicable Diseases and Transitioning Health System in the Democratic People's Republic of Korea during COVID-19 Lockdown

Dear Editor,

Thank you for your constructive suggestions regarding our manuscript, entitled “Non-communicable Diseases and Transitioning Health System in the Democratic People's Republic of Korea during COVID-19 Lockdown”. We have revised the manuscript according to the recommendations provided by the reviewers, and we believe that this process has improved the manuscript substantially. We are pleased to resubmit our revised version to be considered for publication in the Healthcare.

We have uploaded a version with tracked changes that reflects the modifications incorporated into the manuscript, and a point-by-point response to the reviewers’ comments that details the changes made in response to these comments.

Thank you for your consideration of this paper. Please feel free to contact me if you have any questions related to this manuscript.

Sincerely,

Jiho Cha, MD, PhD

Moon Soul Graduate School of Future Strategy, KAIST,

Daejeon, Republic of Korea; [email protected]

Reviewer 2 Report

Dear Authors,

It is interesting topic. However, it would be valuable to mention sth about similar research conducted in other countries. Therefore, wider literature review is needed. You should also extend the reference list as it is very limited. 

The goal / purpose should be more clearly specified. At the present moment it is too long and in fact it focus on the activity such as reviewing articles, etc from some perspective - (lines 53 -56 "Given the scarcity of available literatures in North Korea, this study conducted a  scoping review of peer review articles and policy related literatures through state-run newspaper to analyze the NCDs in North Korea in particular consideration of health system impacts of COVID-19 pandemic"). So, the goal should be clearly specify not by the activity or method such as reviewing articles etc.

In both introduction and especially in discussion part you should underline what is the importance of you research for theory and practice. What is the theoretical and practical importance of your research ?

It would be also good to present more directions of your research applications. What can be done based on your research ?

Author Response

(The authors gave the same response as above.)

Reviewer 3 Report

Non-communicable Diseases and Transitioning Health System in the Democratic People's Republic of Korea during COVID- 19 Lockdown

What are the main contributions of your work.

We searched a total of 243 papers (117 Medline, 75 Scopus, and 51 Web of Science) that were potentially relevant to the study subject. The authors can mention the date of access.

According to the major causes of death in North Korea in 2017, NCD mortality of cancer and cardiovascular disease were higher than the mortality rate from infectious diseases where infectious diseases accounted for 4.8%, cancer 17.7%, and cardiovascular disease 38.4%. The authors should mention the source of information.

Figure 1. Flow Diagram of Study Selection. It is not clear.

The authors can make the table in explaining all the survey of the study.

There are no challenges, future directions.

The authors can refer Deep learning and medical image processing for coronavirus (COVID-19) pandemic: A survey

An incentive based approach for COVID-19 planning using blockchain technology

Author Response

(The authors gave the same response as above.)

Round 2

Reviewer 1 Report

Please check the numbering of the bibliography

Author Response

Thank you for your review.

We checked the numbering of the bibliography.

Reviewer 2 Report

Dear Authors, 

Generally more comments were taken into account. However the importance and applications of these research could be more extended.

Author Response

We added importance and applications of these research(p6).

Reviewer 3 Report

Paper can be accepted 

Author Response

Thank you for your review.
